# How conformity can lead to polarised social behaviour

**Folco Panizza** [1,2] *, **Alexander Vostroknutov**[3], **Giorgio Coricelli**[4,5]

**1** Molecular Mind Laboratory, IMT School for Advanced Studies Lucca, Italy, **2** Center for Mind/Brain Sciences, University of Trento, Mattarello (TN), Italy, **3** Department of Economics (MPE), Maastricht University, Maastricht, the Netherlands, **4** Department of Economics, University of Southern California, Los Angeles, California, United States of America, **5** LaPsyDÉ, UMR CNRS 8240, La Sorbonne, Paris, France

\* folco.panizza@imtlucca.it

**Data Availability Statement:** Behavioural and computational data, as well as a live version of the code used for the analysesare available via the Open Science Framework (osf.io/p5xq3).

**Funding:** GC acknowledges the financial support of the European Research Council (ERC Consolidator

## Abstract

Learning social behaviour of others strongly influences one's own social attitudes. We compare several distinct explanations of this phenomenon, testing their predictions using computational modelling across four experimental conditions. In the experiment, participants chose repeatedly whether to pay for increasing (prosocial) or decreasing (antisocial) the earnings of an unknown other. Halfway through the task, participants predicted the choices of an extremely prosocial or antisocial agent (either a computer, a single participant, or a group of participants). Our analyses indicate that participants polarise their social attitude mainly due to normative expectations. Specifically, most participants conform to presumed demands by the authority (vertical influence), or because they learn that the observed human agents follow the norm very closely (horizontal influence).

## Author summary

What drives people to extreme acts of generosity? What causes behaviour that is unduly spiteful? This study explored how our social decisions polarise. Participants chose whether to spend money to increase or decrease the earnings of an unknown person. Halfway through this task, they observed another agent playing. The agent took participants' choices to the extremes: if for instance the participant was moderately generous, it spent considerable sums to help the other. Participants conformed regardless of whether the agent was a computer algorithm, a person, or a group of people. We tested several competing explanations of why this happened with the help of cognitive modelling. Our analyses identify two factors behind polarisation: willingness to comply with the experimenter expectations (social desirability), and concern about appropriate behaviour (norm conformity). Our approach provided insight into how social choices are influenced by others, and could be applied in the study of conformity in other types of decisions.

Grant 617629; https://erc.europa.eu/funding/
consolidator-grants). The funders had no role in
study design, data collection and analysis, decision
to publish, or preparation of the manuscript.

## Introduction

Recent years have seen a growing concern with online discourse promoting violence, such as cyber-bullying or hate speech [1]. Increasing exposure to uncivil commenting, besides taking substantial psychological and societal toll [2], is thought to reinforce users' toxic behaviours [3, 4], political polarisation [5], or their perception of political divide [6]. Conversely, viral trends can also lead to pro-social outcomes: learning about others' donation choices increases individuals' willingness to give to charity [7, 8]. Evidence suggests that fund-raising success of charitable initiatives is predicted by how much they are shared by social network users [9], or by how concerted the network structure is [10]. If people's attitude becomes more charitable or more malevolent in these contexts, this is at least partly due to social conformity [11–13].

Insights on the cognitive mechanisms behind anti- and prosocial conformity come from the literature on attitude alignment and preference learning [14, 15]. These studies have spanned a variety of domains such as attractiveness ratings [16], food [17], risk preferences [18–20], moral behaviour [21], effort [20], and inter-temporal decisions [20, 22–25]. At a brain level, learning about others' attitudes or preferences appears to alter the value representation of choices [16, 21, 22, 24] or even reward signals [19], while not necessarily affecting one's private preferences [18]. In addition, features such as choice variability [23] or attitude extremeness [25] seem to be significant predictors of conformity.

Recent studies have also brought attention towards the behavioural aspects of social conformity [8, 26–29]. The work by Dimant and colleagues shows for instance how anti-social models beget a higher degree of conformity compared to pro-social models, and that social proximity to the model is also a strong predictor of conformity.

While this research helps untangling the brain bases and behavioural ramifications of preference conformity, it remains largely unclear why exactly people shift their attitude in the direction of others' behaviour in general, and their social attitude towards other individuals in particular. In this preregistered study (osf.io/th6wp; changes to the original protocol: S1 Methods) we test several competing mechanisms that were proposed as explanations of attitude conformity. We consider five competing hypotheses. The *time-dependence* hypothesis predicts that people change their social attitude even in the absence of any observation. Indeed, there is preliminary evidence that during strategic interactions, participants' behaviour becomes more self-oriented with time [30–33]. The *contagion* hypothesis [19, 34] posits that attitude conformity is the result of some kind of automatic imitation of an agent's behaviour, irrespective of its nature or relationship with the observer. This hypothesis predicts that conformity will occur regardless of whether the observed agent is human or non-human. The *compliance* hypothesis states that participants could change attitude due to the mere presence of an authority, in our case the experimenter [35]. This hypothesis predicts that a portion of participants would change their attitude in any context where they think they are expected to, rather than actually reacting to others' behaviour. The *preference learning* hypothesis [23] posits that people are unsure about what their own preferences are, but they can learn them from the behaviour of others, assuming that the agent's and observer's preferences come from a common distribution. Other people's choices can thus be used to learn how one wants to behave, rather than how one ought to behave. Since preference learning should decrease in the process of learning others' choices, a second prediction of this hypothesis is that participants' behaviour should become more consistent after learning.

The last hypothesis that we test, *norm learning*, states that attitude conformity stems from learning what behaviour is socially appropriate or how much social appropriateness matters in a given context. We conjecture that in many real life situations there is a considerable amount of uncertainty about what constitutes a social norm or how salient it is [36]. Furthermore,

many studies have shown that people have a strong preference to follow norms conditional on others following them as well [37, 38]. Thus, observing other people's behaviour should reveal either information about what others believe is "the right thing to do" or at least how frequent or infrequent deviations from the norm are [33, 39]. This hypothesis makes two separate predictions: that participants conform after changing their beliefs about which norms are in place (*norm uncertainty*), or rather that participants are aware of the existing norms, but conform after learning how strictly the norm is followed (*norm salience*).

To our knowledge of the various literatures surveyed, these five hypotheses exhaust the list of tested explanations of conformity in decision making. Therefore, our approach is to falsify as many hypotheses as we can, and attribute the behaviour to the hypotheses that we cannot reject. To disentangle the predictions of these five hypotheses, we use a series of between-subjects experimental conditions. In all conditions participants play a resource-allocation game where in each round they choose between two money allocations to themselves and another unknown participant. Halfway through the game, participants are asked to predict and learn the choices made by another agent in the same task. Depending on the condition, the agent is either a computer, a previous participant, or a group of previous participants (in the Baseline condition participants do not observe anyone). After the main part of the experiment, we administer another task measuring the normative beliefs of participants.

We use a series of cognitive models of participants' decisions to analyse behaviour in the resource-allocation game. These models link behaviour to the mental processes associated with the different hypotheses. Testing of the competing mechanisms behind social conformity is then performed by comparing social attitudes before and after the manipulation phase, and using additional evidence collected during and after the main task. Model estimation is essential for distinguishing different sources of attitude variability, as for instance participants' own variability in behaviour and attitude changes induced by learning.

## Methods

### Ethics statement

The local Ethical Committee of the University of Trento approved the study and subjects provided written informed consent prior to their inclusion.

### Participants

Participants were recruited through the recruitment system of the Cognitive and Experimental Economics Laboratory (CEEL) at the University of Trento and contacted via e-mail. No particular exclusion criteria were defined, with the only exception that participants should not have taken part in other experiments involving a similar task. Payments were made in cash and varied depending on participants' choices.

To determine the sample size necessary to detect a change in social attitude, we conducted a power analysis using G*Power [40] aiming to obtain.95 power, .05 $\alpha$ probability, and at least a small-to-medium effect size (Cohen's $d = 0.35$ for all tests). This effect size figure was recently proposed as a plausible mean prior for experiments in social psychology [41]. As the original hypotheses were directional (i.e., participants' attitudes shift towards the agent's attitude), tests considered were one-tailed one-sample $t$-tests against constant, and two-tailed two-sample $t$-tests (for post-hoc pairwise comparisons between conditions, uncorrected). Calculation yielded a sample size of 90 participants in order to achieve the required power across all conditions. Due to an unforeseen limit in the size of the recruitment pool however, the samples of the last two conditions in order of acquisition were smaller than this pre-specified size (74 and 66 participants). Unbalance in sample size across experimental conditions is not particularly

concerning as the tests used to compare conditions (see Attitude convergence) are non-parametric and therefore do not require the assumptions typically achieved with samples of similar size such as homoscedasticity.

376 participants (age $M$ = 22, $SD$ = 2, 167 males) took part in the experiment. Data from four participants had to be excluded due to failures in the software, as well as the data from three other participants who already participated in a pilot version of the study. Analyses were thus conducted on 369 participants.

## Resource-allocation game

During each trial of the task, participants observed an allocation of points (1 point = 0.10€) distributed between themselves and an unknown other participant, the recipient. Participants were then asked whether they preferred the current allocation of points or a default allocation (100 points to oneself, 50 points to the recipient, Fig 1B). Participants played the game twice, with different allocations, before and after the manipulation phase. At the end of the experiment, participants were randomly paired, and one participant in each pair was randomly selected: one of the selected participant's decisions was randomly sampled and implemented for payment (i.e. the selected participant earned the points for herself and the non-selected participant received the points for the other).

101 alternative allocations (102 for Baseline condition) and the default allocation were drawn from the set of integer allocations closest to the circumference of radius 50 centred at (50, 50) (Fig 1A). Compared to the default allocation, these alternatives provided less points to the participant, but could in exchange affect the recipient's payoffs: half of the alternative allocations were more advantageous for the recipient than the default allocation ("prosocial" trials), whereas the other half left the recipient worse off ("antisocial" trials). In addition, 9 alternative allocations were more profitable for the participant (making her earn more than 100 points) whereas the recipient gained 50 points, as in the default allocation.

To define the allocations around the circumference, we first considered all integer coordinates within one point tolerance from the circumference (i.e., all values between a circumference of radius 49 and a circumference of radius 51). Second, only points between 112.5˚ and −112.5˚ were included for the analyses; this range excluded allocations that were too extreme (e.g. (15, 15) or (0, 50)). Third, we excluded allocations with more points to oneself than the default option, because gains for the player risk to overshadow the difference in points for the other. Likewise, we excluded allocations with the same points to the other as the default option. Finally, we eliminated allocations that give more than 100 or less than 0 to the other player. This procedure yielded 406 allocations in total; these were then divided in four subsets, two of 101 and two of 102 trials, all evenly distributed around the arc of the circle. The two subsets of 102 trials were used in the Baseline condition (in the choice parts of the task), whereas the remaining subsets of 101 trials were used in the other conditions (S1 File contains a full list of trials).

The use of the circumference as a way to select trials was based on two considerations. First, the circumference has been used in the previous literature as a measure of social orientation ([42] is the seminal paper), and should yield comparable results. Second, many models have been adopted in the literature to describe how people value options in social decision-making (e.g., [43–46]); by using allocations around a circumference, most of these models make the same predictions. Agreement in model predictions allowed us to use a very simple utility model (Eq 1), with only one variable defining the attitude of the decision-maker. A simple model greatly simplifies computations and thus helps testing our cognitive predictions concerning attitude conformity and choice consistency.

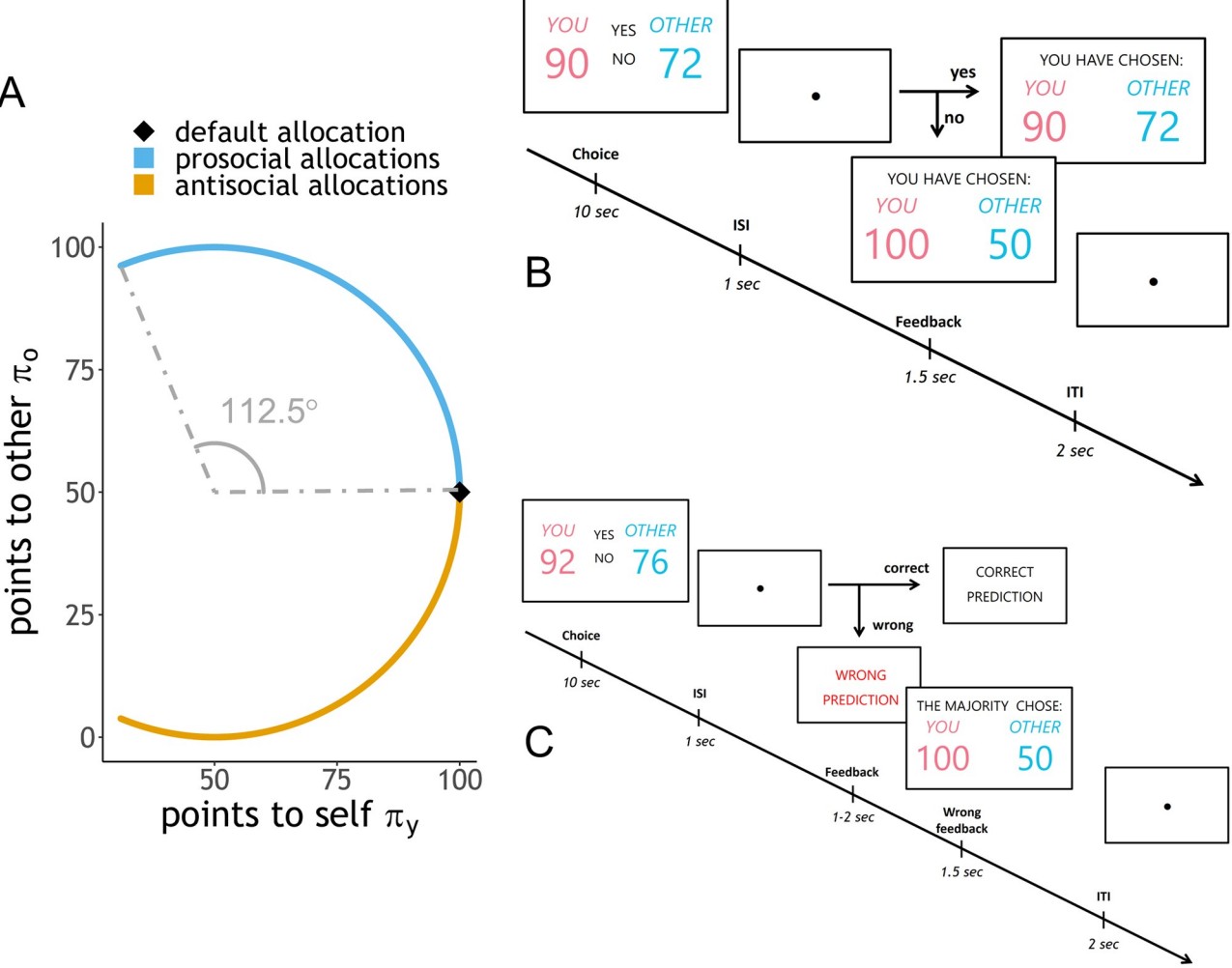

**Fig 1. Trials in the experiment.** The complete list of trials is available at osf.io/th6wp. **A**: Participants chose between a default allocation (black rhombus) and an alternative allocation, which could be either prosocial (light blue) or antisocial (orange). Allocations were limited to a certain arc of the circumference and to a certain range, with the exception of 9 allocations giving more points to the participant (not shown in the figure but included in S1 File). **B**: resource-allocation game. Participants observed the current alternative allocation and had a maximum of 10 seconds to respond. Decision cues ('yes'/'no') indicated which button to press for each decision (up/down arrows). Points for self and for the other were colour-coded (points for self: red; points for the other: blue) and were presented on the left and on the right of the decision cues. Both cues and points (self/other) switched position randomly across trials. If participants did not answer within 10 seconds, the trial ended, and they were automatically assigned the default allocation. Unanswered trials were considered missing data. After the decision, an inter-stimulus interval of 1 second divided the decision and the feedback. Feedback lasted for 1.5 seconds and displayed the allocation preferred by the participant. The trial ended with an inter-trial interval of 2 seconds. **C**: manipulation phase. Participants were presented with an alternative allocation, and indicated whether they believed the agent preferred the alternative over the default allocation ('yes') or not ('no'). The choice could be made within 10 seconds, after which it would no longer be valid. After an inter-stimulus interval of 1 second, participants received feedback about their answer. If the prediction was correct, the feedback message 'correct prediction' appeared on the screen for approximately 1.5 seconds (minimum 1, maximum 2). If the prediction was wrong or not given in time, a similar feedback message ('wrong prediction' or 'no answer') appeared on the screen for about the same time, followed by the actual choice of the agent, lasting 1.5 seconds. The name of the agent varied between conditions (Group condition: 'majority'; Individual condition: 'participant'; Computer condition: 'computer'). After the feedback, the trial ended with an inter-trial interval of 2 seconds.

To estimate attitude towards others, we assume that participants can attribute to each allocation of points a unique subjective value. Value of an allocation is computed according to Eq 1:

$$V(\pi_y, \pi_o) = \pi_y + \tan(\alpha) \cdot \pi_o,$$ (1)

where $\pi_y$ and $\pi_o$ are respectively the amount of points for oneself (you) and for the other, and $\alpha$ represents the "social value orientation" or social attitude of the participant [47, 48]. The attitude defines how much and in what way the amount of points for the other plays a role in the participant's decisions; in fact, $\tan(\alpha)$ represents how much one point for the other person is worth in terms of one's own points (e.g., when $\alpha = 30°$ one point for the other is roughly equal to 0.58 points for oneself). If $\alpha$ is positive (negative), then the higher (smaller) amount of points for the other makes the player better off. A participant with a positive $\alpha$ is said to be *prosocial*, whereas a participant with a negative $\alpha$ is said to be *antisocial*.

Social attitude—together with other parameters relevant to the decision process—is estimated twice, for choices before ($\alpha_{before}$) and choices after ($\alpha_{after}$) the manipulation phase. A separate estimation allows measuring any change in attitude that ensues from the manipulation phase (Cognitive modelling and Model comparison).

## Manipulation phase

In the Computer, Individual, and Group conditions, after the first part of the resource-allocation game, participants were asked to predict the choices of an agent in a different set of alternative allocations (Fig 1C). Participants played 63 trials of the manipulation phase in all conditions except Baseline. Correct predictions were incentivised to ensure that participants paid attention to the task. Participants received immediate feedback after each prediction, so that they could correctly learn about the agent's attitude.

The attitude of the observed agent ($\alpha_{obs}$) was controlled experimentally unbeknownst to participants. Specifically, if participants displayed a prosocial attitude in the first part of the game ($\alpha_{before} > 0$), they observed an agent with an extremely prosocial attitude ($\alpha_{obs} \approx 45°$, one point for the other equals one point for the self); and vice versa: if participants displayed an antisocial attitude ($\alpha_{before} < 0$), they observed an agent with an extremely antisocial attitude ($\alpha_{obs} \approx -45°$, one point for the other equals negative one point for the self).

The behaviour of the observed agents was based on real choices of participants in the Baseline condition taken from either the first or second part of the resource-allocation game. In the Individual condition, the agent was a single previous participant, with an estimated $\alpha$ value close to $45°$ (prosocial agent) or $-45°$ (antisocial agent). In the Group condition, the agent consisted of a group of five previous participants (the size of the group was not mentioned in the instructions). The choices shown to participants referred to the allocation preferred by the majority of the group, that is the modal response. Lastly, in the Computer condition, participants were told that the agent was a computer selecting options according to a predefined criterion. The criterion of the computer agent was in fact to choose exactly as the group in the Group condition.

We chose to display extreme agents in order to distinguish attitude conformity from a gradual increase in selfishness that was observed by [30] and [32]. Hence, if participants with a moderate attitude conformed to the agent's attitude, this change could not be attributed to an increase in selfishness. In addition, if participants did learn about decision makers who were less extreme than themselves, the attitude change would push them in the same direction as the regression to the mean. Any movement of the attitude away from the mean cannot then be obfuscated by this effect. In addition to experimentally manipulating agents' attitudes, we also carefully calibrated the consistency of choices. This procedure ensured that agents displayed consistent patterns of choice, so that their attitudes could be easily predicted by participants.

We calibrated $\alpha_{obs}$ based on the participant's attitude in the first part of the resource-allocation game, before the manipulation phase ($\alpha_{before}$). Since we could not tell *a priori* which cognitive model would fit participants' data best (Cognitive modelling), we determined whether

participants had an $\alpha_{\text{before}}$ greater or less than zero (prosocial or antisocial) using the following formula 2:

$$\sum_t \mathbb{I}_{t,A} \cdot \text{atan}\left(\frac{\pi_{ot} - 50}{\sqrt{(\pi_{ot} - 50)^2 + (\pi_{yt} - 50)^2} + \pi_{yt} - 50}\right), \quad (2)$$

where $\pi_{yt}$ and $\pi_{ot}$ are the points in the alternative allocation for self (you) and the other in trial $t$ of the resource-allocation game, and $\mathbb{I}_{t,A}$ is an indicator variable that equals 1 when the participant preferred the alternative allocation in trial $t$ and 0 when the participant preferred the default allocation in trial $t$.

The dependent variable that we use to measure conformity is *attitude convergence*, denoted by $\delta_{\text{diff}}$. To compute its value, we use Eq 3 (coincidentally similar to the Contagion Gap of [26]):

$$\delta_{\text{diff}} = \delta_{\text{before}} - \delta_{\text{after}} = |\alpha_{\text{before}} - \alpha_{\text{obs}}| - |\alpha_{\text{after}} - \alpha_{\text{obs}}|, \quad (3)$$

where $\delta_{\text{before}}$ and $\delta_{\text{after}}$ are the distances between the attitude of the observed agent $\alpha_{\text{obs}}$ and participant's attitude estimated respectively before and after the manipulation phase. In order to have comparable results with the other conditions, we use this measure also for Baseline participants as if they were predicting choices of an agent from the Group or the Computer condition. As an exercise of parameter recovery for this variable, see S2 Analyses.

We have chosen $\delta_{\text{diff}}$ as a measure of attitude conformity because it has two critical advantages over previous measures used in the literature [22, 24]. First, $\delta_{\text{diff}}$ depends on the original distance from the model: if a participant's starting attitude is very close to that of the observed agent, then $\delta_{\text{diff}}$ can only be small. As this is a conservative measure, it prevents close participants from biasing the estimate at sample level. Second, by taking into account the attitude distances from $\alpha_{\text{obs}}$ of both $\alpha_{\text{before}}$ and $\alpha_{\text{after}}$, $\delta_{\text{diff}}$ differentiates between participants who shift attitude closer to the agent, and those who overshoot and become more extreme than the agent. It is indispensable to distinguish between these two types of attitude change, as the hypotheses that we test–with the exception of time-dependence–are concerned only with the former kind (moving closer to the agent).

Attitude convergence unambiguously predicts participants' attitude to converge towards the observed agent's attitude, hence this measure also penalises attitude changes that lead the participant to become *more extreme* than the observed agent. As a robustness check, we show that all the main results hold using an alternative measure that accounts for polarisation (see S6 Analyses):

$$\delta_{\alpha} = \text{sgn}\,\alpha_{\text{obs}}(\alpha_{\text{after}} - \alpha_{\text{before}}), \quad (4)$$

Predictions regarding attitude convergence for each of the five hypotheses are summarised in the left part of Table 1. Notice that the predictions of the time-dependence hypothesis have an opposite direction compared to all other hypotheses. Moreover, the two versions of the norm learning hypothesis (norm uncertainty and norm salience) make no specific prediction about attitude change in the Individual condition.

## Other measures

While attitude convergence is the main measure that we use to distinguish among the hypotheses, we also need to test ancillary predictions that these hypotheses make to distinguish between the contagion and compliance hypotheses, and between the preference learning and

**Table 1. The predictions of the five hypotheses.** "↑" refers to increasing extremeness of the attitudes. "–" means no change predicted. "↓" refers to the shift towards selfishness.

| Hypotheses | Conditions | | | | Other Measures |
|---|---|---|---|---|---|
| | Baseline | Computer | Individual | Group | |
| Time-dependence | ↓ | ↓ | ↓ | ↓ | |
| Contagion | – | ↑ | ↑ | ↑ | |
| Compliance | – | ↑ | ↑ | ↑ | Compliance Index (only ≥25%) |
| Preference learning | – | – | ↑ | ↑ | Consistency increase (in human conditions) |
| Norm uncertainty | – | – | – /↑ | ↑ | Different norms (human vs. computer) |
| Norm salience | – | – | – /↑ | ↑ | Same norms (human and computer) |

norm learning hypotheses. For this purpose, we adopt a series of additional measures. The right-hand side of Table 1 summarises the related predictions.

**Compliance.** The contagion and compliance hypotheses make identical predictions in terms of attitude change. To distinguish between them, we assess participants' tendency to comply with the experimenter's expectations [49, 50]. If the contagion hypothesis is true, we should observe attitude convergence in the Computer condition even after controlling for participants' compliance tendencies. If instead attitude convergence in the Computer condition depends on compliance tendency, this result should support the compliance hypothesis.

Compliance to experimenter demand in standard Dictator Games has been associated with an increase in prosocial behaviour (see for instance [51]). In the resource-allocation game, however, participants can make both prosocial and antisocial decisions, making prosocial behaviour a less obvious choice to appease the experimenter [52]. Moreover, such demand by the experimenter is explicitly ruled out in the instructions, where we specify that we do not expect any particular behaviour, neither prosocial nor antisocial. Evidence for what might constitute compliance in the resource-allocation game comes from an experiment adopting a similar paradigm [53]. This study suggests that when presented with conflicting choices during a task, such as behaving prosocially and antisocially, complying participants think they should demonstrate *both* types of behaviour to meet the experimenter's expectations, even if these choices yield paradoxical outcomes. Critically, such pattern of behaviour has been associated with compliance with experimenter expectations and a personality index measuring social desirability [53, 54]. If a participant displays such behaviour in the resource-allocation game, then it is plausible that authority compliance—rather than conformity to the observed agent—explains her attitude change. To measure authority compliance, we consider separately the proportion of prosocial alternatives and the proportion of antisocial alternatives chosen over the default allocation: we define our index of compliance as the smallest of these two numbers in percentage terms.

To distinguish between compliant and non-compliant participants, we use a preregistered threshold set to 25% (osf.io/th6wp; see S1 Table for the robustness of results adopting different thresholds). In other words, a participant is said to be compliant if she chose both prosocial and antisocial alternatives at least once out of every four choices made. We use the compliance index to test attitude convergence in the Computer condition. If compliant participants change attitude but non-compliant participants do not, we interpret this evidence in favour of the compliance hypothesis. If instead participants change attitude regardless of the compliance index, we interpret this evidence in favour of the contagion hypothesis. As an exploratory analysis, we also treat compliance index as a continuous variable, to test whether attitude convergence is linearly associated with compliance.

**Preference learning.** The preference learning hypothesis predicts that participants change their attitude because they learn their own social preferences from others. Since learning in this case should reduce participants' uncertainty about how they want to behave, we should observe a corresponding increase in choice consistency after the manipulation phase in human (Individual, Group) relative to non-human (Baseline, Computer) conditions. In other words, consistency (variability) between choices should reflect how certain (uncertain) a person is about her social attitude, and consistency should increase after learning about the preferences of others.

While increased consistency is a precondition for preference learning, participants might also become more consistent if the norm learning hypothesis is true. Contrary to preference learning, however, norm learning does not exclude that participants already follow a social norm even before observing the agent. If this is the case, the information obtained from the observed agent might add knowledge about the norm without necessarily increasing choice consistency. Therefore, if our analyses fail to confirm a differential increase in consistency between human and non-human conditions, we will interpret this evidence as being against the preference learning hypothesis but not against the norm learning hypothesis.

We can test for changes in consistency by looking at the cognitive model used to understand participants' choices, and in particular at the parameter representing variability in participants' choices. Depending on the winning cognitive model, this parameter is either $\tau$ or $\sigma$ (see see Variability parameters $\tau$ vs. $\sigma$): a small $\tau$ ($\sigma$) corresponds to very consistent choices and vice versa. Hence, to test the preference learning hypothesis we measure whether participants' variability decreases after the manipulation phase (e.g., for $\sigma$, $\sigma_{after} < \sigma_{before}$). If participants do become more consistent after the manipulation, we will further test whether consistency increase is significantly different between conditions, and particularly between the human and non-human conditions.

**Norm following.** The norm learning hypothesis assumes that participants' behaviour is influenced by beliefs about what constitutes a socially appropriate or inappropriate action. Accordingly, we should expect that prosocial and antisocial participants have different beliefs about what choices are considered appropriate in the resource-allocation game. To measure appropriateness perception, participants in the Computer, Individual, and Group conditions completed the norm elicitation task [55] at the end of the experiment. In this task, participants rated on a 4-point Likert scale the degree of social appropriateness of choosing the alternative allocation over the default option in a selection of choices from the resource-allocation game. If one of these ratings, randomly chosen, matched that of the majority of other participants in the experimental session, then the participant was rewarded with 3.00€. This procedure ensured that participants reported their true beliefs about what the majority thought was socially appropriate, namely what constituted a social norm. Using the norm elicitation task, we test whether prosocial and antisocial participants have different perceptions of the social norms in the game. This is done in the Computer condition where no social information can be acquired from the agent. We expect that any difference in appropriateness ratings is due to participants' original beliefs before the task. If prosocial and antisocial participants do indeed report different normative beliefs, this could explain their differences in social attitudes, in accordance to the norm learning hypothesis.

**Norm uncertainty and norm salience.** If the results of the elicitation task support the hypothesis of norm learning, we also use appropriateness ratings to explore what kind of information participants learn about the norm. Indeed, observing a social agent allows learning distinct features of a social norm. First, if participants are uncertain about what is appropriate and what is not, observation can provide useful information about the norm itself (*norm uncertainty*). If there is norm uncertainty, we expect observation of a human agent with very

prosocial or antisocial behaviour to polarise the perception of what constitutes a right or wrong choice. Polarisation in turn should lead to more extreme appropriateness ratings in the Group and Individual conditions than in the Computer condition, where participants simply predict the choice patterns of a computer (and therefore learn nothing about social norms).

In addition to learning about the norm itself, observing the behaviour of others reveals whether a norm is actually followed or not (*norm salience*). The norm elicitation task, however, is not designed to measure norm salience, and we are unaware of any other task that could possibly elicit this feature of a norm. We therefore test for norm uncertainty by computing differences in appropriateness ratings between the Computer and human conditions, for prosocial and antisocial participants separately. If the ratings differ between the conditions, we interpret this as evidence of norm uncertainty. If ratings do not differ between conditions, we conjecture that norm learning occurred through a change in norm salience.

## Cognitive modelling

**Bias parameter $\kappa$.**   We associate participants' choices to their social attitude via Eq 1. Yet this estimate or that of other parameters could be biased by the participant's tendency to comply to authority (see Other measures). To control for compliance, we allow for the possibility that participants prefer an alternative allocation even when it should be on a par with the default allocation, or vice versa. To represent this change in subjective value of the alternative allocation, we define for each participant a bias parameter $\kappa$ 5:

$$V(D) = V(100, 50) = 100 + \tan\alpha \cdot 50 - \kappa, \tag{5}$$

where $\kappa$ is equivalent to an amount of penalty or bonus points for the default allocation: The higher (lower) $\kappa$ is, the higher (lower) the propensity to choose the alternative over the default allocation. In other words parameter $\kappa$ captures the bias, unexplained by other parameters, according to which participants choose between the alternative allocation and the default allocation *as if* the default allocation was *missing* or having *additional* $\kappa$ points. Although we acknowledge that this parameter could capture phenomena other than authority compliance, we were unable to provide any other interpretation of $\kappa$. In addition, $\kappa$ and the original compliance index strongly correlate (see Cognitive modelling), suggesting a common variance between the two measures.

**Variability parameters $\tau$ vs. $\sigma$.**   During the allocation game, participants might show variability in the way they choose, such as being more or less prosocial (or antisocial) from choice to choice. Not accounting for this variability within each part of the game (before or after prediction) could bias the estimation of the change in attitude due to the manipulation. To estimate choice variability, we compare two types of cognitive models that also give different interpretations about the nature of social attitude.

The first model type, Stable Attitude, assumes that attitudes are a stable personal trait, and that any variability in participants' choices is due to cognitive mistakes when comparing different options. If for instance a person occasionally shows a more prosocial (or more antisocial) attitude than usual, this fluctuation is interpreted by the model as a miscalculation on how to behave. Comparisons errors are modelled through the parameter $\tau$: the smaller (larger) $\tau$ is, the higher (lower) the probability of choosing consistently with one's own attitude. Stable Attitude models compare alternatives using a softmax function (Eq 6, [56]):

$$\Lambda(\Pr(D = 1)) = \frac{V_D - V_A}{\tau}, \tag{6}$$

where $\Lambda$ is the logit link function, $\Pr(D = 1)$ is the probability of choosing the default

allocation, $V_D$ and $V_A$ are the estimated values for the default and alternative allocations, as in Eqs 1 and 5.

The second model type, Variable Attitude, assumes instead that attitude is a variable mental state. If for instance people behave more or less prosocially, this is interpreted as a natural fluctuation of attitude. Participants' choices are modelled using random preference [57–59]: every time the participant has to make a decision, her social attitude $\alpha$ is sampled from a normal distribution with centre $\mu$ and standard deviation $\sigma$. The parameter $\sigma$ represents variability in the way participants behave: the smaller (larger) $\sigma$ is, the more (less) consistent the participant will be across her choices. The model is defined as 7:

$$
\begin{cases}
\dfrac{T_\alpha - \mu}{\sigma}, & \text{if } \pi_o > 50 \\[2ex]
\dfrac{\mu - T_\alpha}{\sigma}, & \text{if } \pi_o < 50
\end{cases},
\tag{7}
$$

where $\Phi$ is the probit link function, $\Pr(D = 1)$ is the probability of choosing the default allocation, and threshold $T_\alpha$ is the value of $\alpha$ for which the default and alternative allocations have equal subjective value ($V_D = V_A$, Eq 8):

$$
T_\alpha = \operatorname{atan}\left(\frac{\pi_y - 100 + \kappa}{50 - \pi_o}\right),
\tag{8}
$$

If the sampled $\alpha > T_\alpha$, an allocation is preferred and consequently taken, otherwise the other option is chosen.

**Error parameter $\varepsilon$.** The error parameter $\varepsilon$ defines the probability with which participants make a mistake in implementing their choice (e.g., mistyping or inattention). The probability of choosing the default allocation is expressed as Eq 9:

$$
\Pr(D = 1) = (1 - \varepsilon) \cdot \Pr_{\text{model}} + \varepsilon \cdot \frac{1}{2},
\tag{9}
$$

where $\Pr_{\text{model}}$ represents the probability of choosing the default allocation according to the model under consideration (Eq 6 for Stable Attitude or Eq 7 for Variable Attitude). The error parameter thus allows to assume that participants' answers are a mixture between model-based choices and random errors.

**Model estimation.** We estimate three versions of each model type. In the full version of a model, all parameters are estimated twice, before and after the manipulation phase. A second, simpler version of the models assumes that social attitude $\alpha$ is fixed for the whole task, as if it could not change with the manipulation; $\alpha$ is thus estimated only once across all choices. In the third version of the models instead, it is the variability parameter ($\sigma$ for Variable Attitude or $\tau$ for Stable Attitude) to be estimated once for the whole task, as if participants could not get more or less self-consistent in their choices after the manipulation phase. Models thus vary based on two factors: 2 (Stable Attitude / Variable Attitude) × 3 (fixed attitude / fixed variability / both vary). Consequently, we estimate and compare 6 unique models.

Models are estimated in JAGS [60] using the rjags [61] and R2jags [62] packages. Parameters are fitted using Hierarchical Bayesian Analysis (HBA, [63]) on two levels: a sample level (by subject) and a subject level (by time: before/after prediction). For each model, we ran 4 Markov chains for 100,000 iterations, with a burn-in period of 5,000 iterations and a thinning rate of 4. The model with the lowest Deviance Information Criterion (DIC) is selected and used for the statistical analyses. We use the maximum a posteriori (MAP) estimate to derive the most likely value for each parameter, including attitude convergence $\delta_{\text{diff}}$.

## Results

### Cognitive modelling

**Model comparison.** The cognitive model that describes participants' behaviour best is the full version of the Variable Attitude model in which both $\alpha$ and $\sigma$ vary before/after the manipulation phase ($DIC_{VA}$ = 32997.4, Fig 2). Model comparison thus suggests that both attitude and attitude variability change after the manipulation phase, and that $\alpha$ varies across trials rather than being stable. This latter finding is further supported by the generally lower DIC values of Variable Attitude models as compared to all Stable Attitude models.

Despite our original attitude categorisation of participants was agnostic with respect to the winning cognitive model, 346 participants out of 369 (93.8%) were classified in the same categories when using $\alpha_{\text{before}}$ estimates from the Variable Attitude model. Mismatch affects only participants with a moderate social attitude (mean $|\alpha_{\text{before}}|$ = 2.12˚, max = 12.91˚, $N_{\text{Baseline}}$ = 10, $N_{\text{Computer}}$ = 5, $N_{\text{Individual}}$ = 3, $N_{\text{Group}}$ = 5).

**Compliance and $\kappa$ relation.** We test whether the bias parameter $\kappa_{\text{before}}$ estimated before the manipulation phase correlates with the compliance index. If correct, the bias parameter could be then used as an improved measure of authority compliance, in that it could integrate compliance effects directly within the computation of the decision process, and improve the estimates of other parameters. Splitting participants below and above the compliance threshold, we observe that the two groups have significantly different estimated values of $\kappa_{\text{before}}$ (Wilcoxon rank-sum test with continuity correction, $\log(V)$ = 7.65, $p <$.001, $r$ = .51[.42, .61]), with participants below threshold with average $\kappa_{\text{before}}$ = 1.14[0.85, 1.43] and participants above threshold with average $\kappa_{\text{before}}$ = 11.48[8.78, 14.19]. We then measure the association between the two measures using a Spearman's rank correlation, and find a significant association ($\rho$ = .62[.55, .67], $p <$.001). These results support the hypothesis that the bias parameter $\kappa$ in our model also captures a significant portion of the effect of authority on participants.

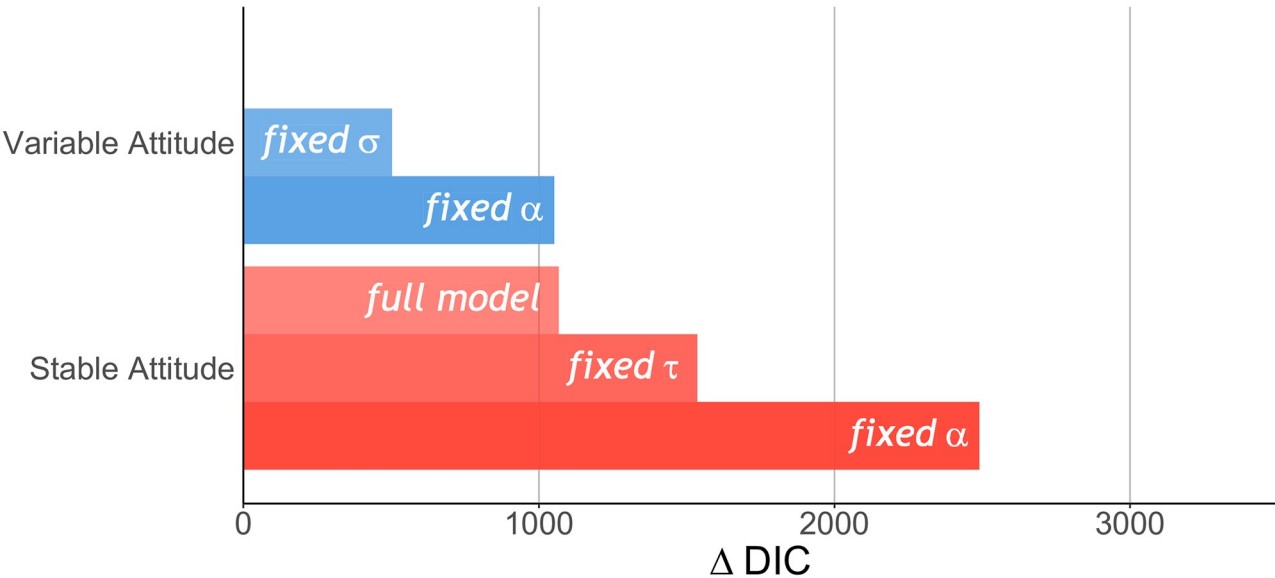

**Fig 2. Model comparison.** The difference $\Delta DIC$ between the Deviance Information Criterion (DIC) of a model and the full version of the winning Variable Attribute model. The Variable Attitude models are in blue, and the Stable Attitude models are in red.

## Attitude convergence

**Preliminary results.**   Based on choices before the manipulation phase, 75% (25%) of participants were categorised as having a prosocial (antisocial) attitude. When presented with prosocial alternatives, prosocial participants chose them over the default allocation 40% of the time, while antisocial participants did the same with antisocial alternatives around 54% of the time. According to our model estimates, mean social attitude before prediction $\alpha_{before}$ was 20˚ ($SD = 14$˚) for prosocial and −22˚ ($SD = 20$˚) for antisocial participants (see panel A of S1 Fig for a comparison across experimental conditions). However, for a significant portion of participants (21% of the sample), 1 point for the other player was worth less than one tenth of a point for oneself ($-5$˚ $< \alpha_{before} < 5$˚), meaning that many participants showed a moderate, if not selfish, social attitude.

In the manipulation phase, prediction accuracy was relatively high: the average number of correct predictions in the last 20 trials was 18.6 ($SD = 2.2$; Computer: 19.1, $SD = 1.8$, Individual: 17.9, $SD = 2.8$, Group: 18.7, $SD = 1.8$). This result suggests that participants had successfully learned the attitude of the observed agent.

**Time dependence.**   If the time-dependence hypothesis is true, participants should become more selfish in all experimental conditions. We first test whether participants chose more often the default, selfish allocation in the second part of the resource-allocation game (after and despite the manipulation phase) than in the first part. Contrary to this prediction, prosocial (antisocial) participants chose *more* prosocial (antisocial) alternatives when learning about the agent's choices (Baseline (no agent): -4.0%, Computer: +0.7%, Individual: +5.3%, Group: +6.9%; S5 Analyses).

These results are mirrored by changes in social attitude, where we find that $\alpha_{after}$ was on average more polarised than $\alpha_{before}$ (Baseline: 0˚, Computer: +4˚, Individual: +7˚, Group: +5˚). Bayes Factor analyses suggest that the data are overwhelmingly more likely under the alternative hypothesis ($H_1$: $\delta_\alpha \neq 0$) than under the null hypothesis (all $BF_{10} > 100$) with the exception of Baseline, where there is substantial to strong evidence in favour of the null ($BF_{01} = 10.2$; see S5 Analyses for a full report of the analyses). These results suggest that time dependence is not present even in the Baseline condition, indicating that this mechanism is not an important factor at play.

**Contagion.**   If the contagion hypothesis is true, attitude convergence $\delta_{diff}$ should be significant and positive in all conditions excluding Baseline. Given that the normality assumption does not hold (Shapiro-Wilk test, all $p < .001$), we test this prediction using a one-tailed Wilcoxon signed-rank test. Indeed, participants, in all conditions except Baseline shifted attitude towards that of the observed agent (Fig 3; Baseline: $\log(V) = 8.40$, $p = .457$, $\delta_{diff} = -1$˚$[-2$˚, $0$˚], $r = .01[-.17, .18]$, $BF_{0+} = 12.32$; Computer: $\log(V) = 7.64$, $p < .001$, $\delta_{diff} = 4$˚$[2$˚, $6$˚], $r = .43[.23, .63]$, $BF_{+0} = 786.20$; Individual: $\log(V) = 7.51$, $p < .001$, $\delta_{diff} = 6$˚$[3$˚, $8$˚], $r = .57[.41, .75]$, $BF_{+0} > 10000$; Group: $\log(V) = 8.28$, $p < .001$, $\delta_{diff} = 5$˚$[4$˚, $7$˚], $r = .58[.43, .72]$, $BF_{+0} > 10000$).

We also measure the difference in convergence across conditions. A Kruskal-Wallis test reveals that there is a significant effect of condition on attitude convergence ($\chi^2(3) = 42.22$, $p < .001$, $\varepsilon^2 = .11[.07, .19]$). Post-hoc pairwise comparisons reveal that attitude convergence differs between the Baseline and Group conditions ($z = 5.87$, $p < .001$, $r = -.39[-.51, -.27]$, $BF_{10} > 10000$), between the Baseline and Individual conditions ($z = 4.68$, $p < .001$, $r = -.33[-.46, -.17]$, $BF_{10} = 717.97$), and between the Baseline and Computer conditions ($z = 3.56$, $p = .005$, $r = -.24[-.38, -.10]$, $BF_{10} = 33.55$). Bayes Factor analyses additionally suggest no difference in attitude convergence between the Group and Individual conditions ($BF_{01} = 5.78$, substantial evidence).

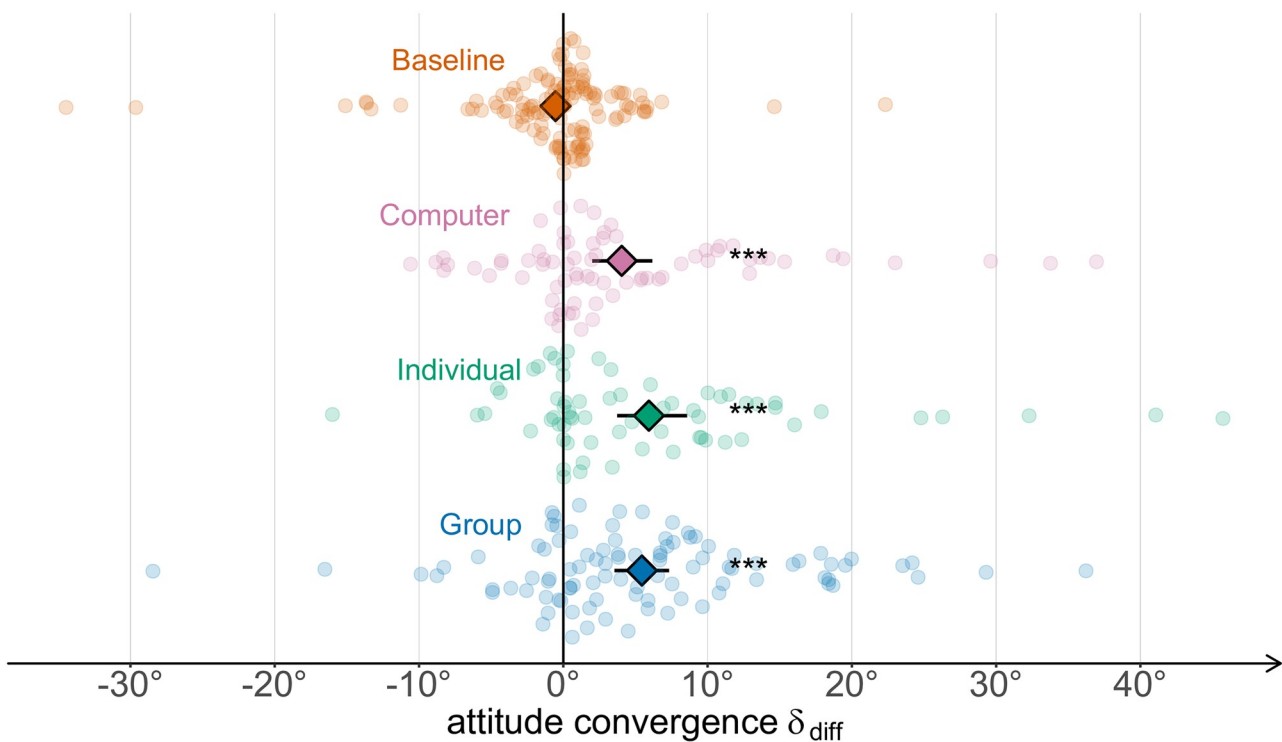

**Fig 3. Mean attitude convergence by condition.** Error bars indicate $t$-adjusted, 95% Gaussian confidence intervals. *: $p < .05$; **: $p < .01$; ***: $p < .001$.

**Compliance.** The above results are compatible with the contagion hypothesis, but also with the compliance hypothesis. To study the influence of the experimenter on attitude convergence, we categorise participants using the compliance index. One participant did not respond in any antisocial trial before manipulation: without these responses we could not compute a compliance index and the participant was consequently excluded for analyses about compliance. We find that 63 participants (Baseline: 23, Computer: 13, Individual: 16, Group: 11), around 17% of the sample, are above the 25% preregistered threshold (Fig 4A). We thus focus on the remaining participants below threshold ($N = 305$). While the results in the Group and Individual conditions hold, attitude convergence in the Computer condition is still significant but weakened (Baseline: $\log(V) = 8.06$, $p = .323$, $\delta_{\text{diff}} = 0°[-1°, 1°]$, $r = .04[-.15, .23]$, $\text{BF}_{0+} = 7.16$, $n_{\text{obs}} = 109$; Computer: $\log(V) = 7.11$, $p = .015$, $\delta_{\text{diff}} = 3°[0°, 5°]$, $r = .30[.06, .53]$, $\text{BF}_{+0} = 6.27$, $n_{\text{obs}} = 60$; Individual: $\log(V) = 6.96$, $p < .001$, $\delta_{\text{diff}} = 5°[2°, 7°]$, $r = .57[.36, .77]$, $\text{BF}_{+0} = 958.61$, $n_{\text{obs}} = 50$; Group: $\log(V) = 8.02$, $p < .001$, $\delta_{\text{diff}} = 5°[3°, 7°]$, $r = .55[.39, .71]$, $\text{BF}_{+0} = 6652.22$, $n_{\text{obs}} = 86$).

In support of this observation, we find that Baseline and Computer conditions are no more significantly different ($z = 2.06$, $p = .058$, $r = -.15[-.30, .01]$, $\text{BF}_{10} = 1.08$; Kruskal-Wallis test: $\chi^2(3) = 31.72$, $p < .001$, $\varepsilon^2 = .10[.05, .19]$, $n_{\text{obs}} = 305$). Instead, Baseline and Group conditions are still significantly different ($z = 5.20$, $p < .001$, $r = -.37[-.51, -.24]$, $\text{BF}_{10} = 2117.44$), and so are Baseline and Individual conditions ($z = 3.90$, $p = .001$, $r = -.31[-.45, -.15]$, $\text{BF}_{10} = 54.87$).

To explore the relation between compliance and attitude convergence, we run a robust linear regression [64] with attitude convergence as dependent variable, and with experimental condition and the interaction between compliance and experimental condition as predictor variables (Fig 5A). We compare the deviance of the regression against a simpler model using only experimental condition as an independent variable: the full model has a better fit on the

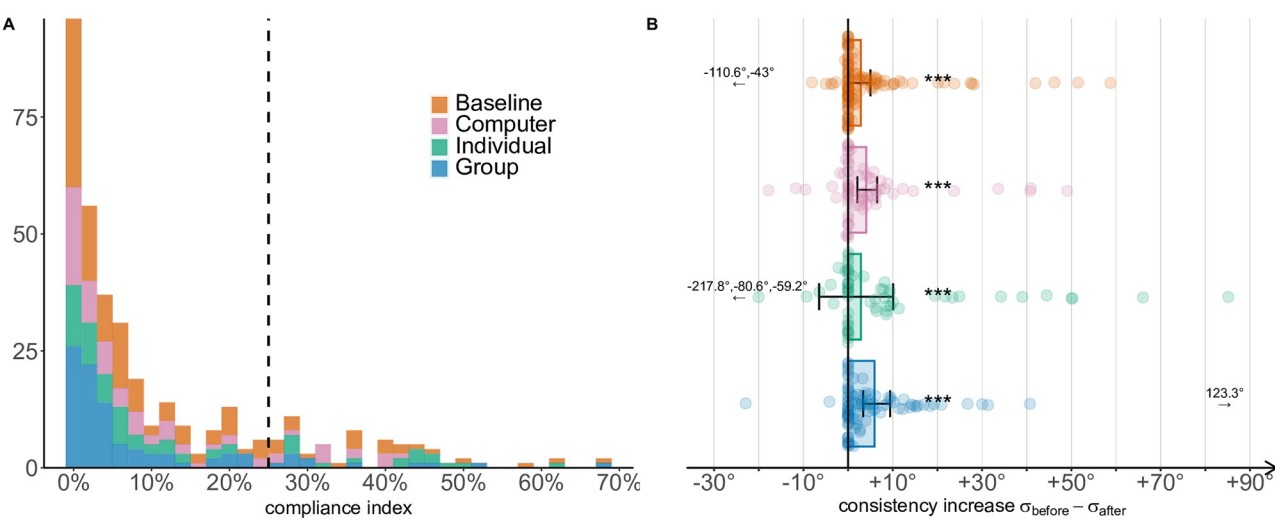

**Fig 4. Compliance index and consistency increase.** A: Distribution of the compliance index across all participants. The vertical line shows the threshold value (25%) beyond which a participant is considered to be susceptible to authority compliance. B: Consistency increase across conditions. Participants become more consistent after the manipulation phase ($\sigma_{after} < \sigma_{before}$), but the increase is not significantly different across conditions. Error bars indicate $t$-adjusted, 95% Gaussian confidence intervals. Arrows indicate outliers. *: $p < .05$; **: $p < .01$; ***: $p < .001$.

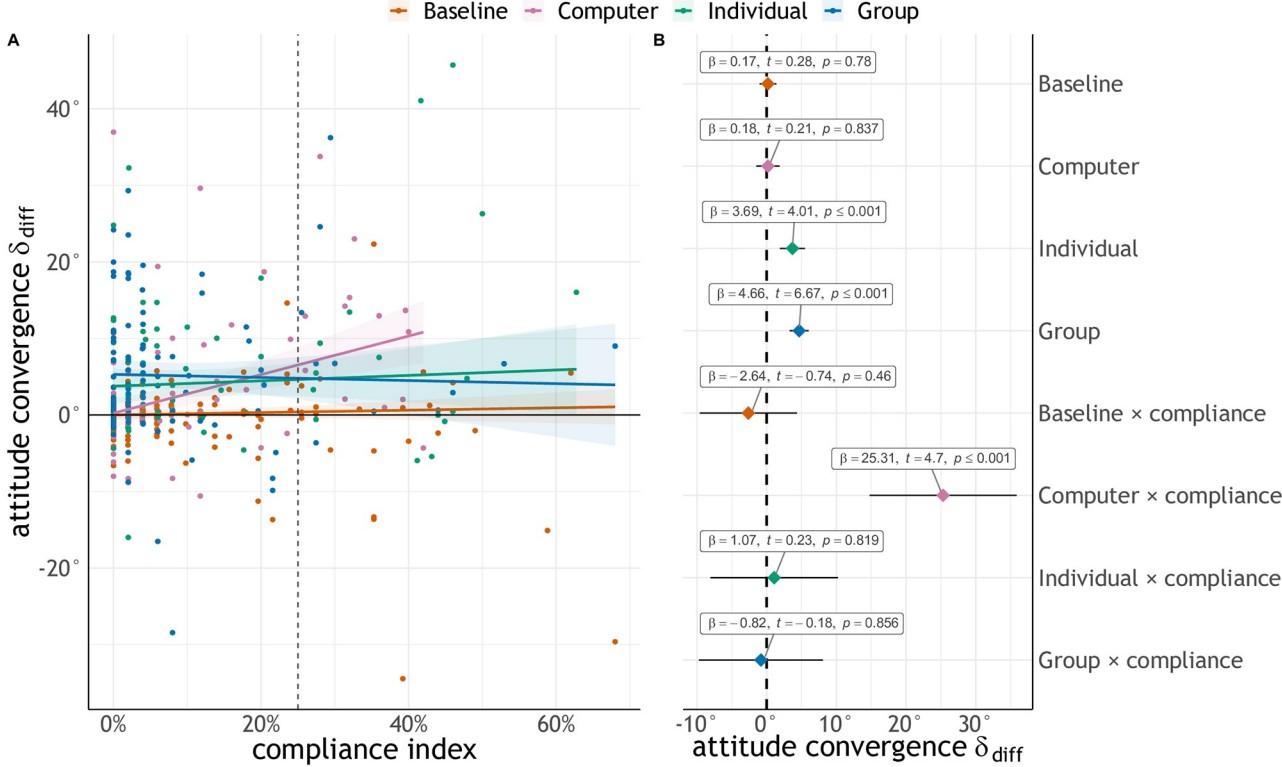

**Fig 5. Robust regression results.** A: Robust regression on attitude convergence with experimental condition and the interaction between compliance and experimental condition as predictor variables. Shaded areas indicate 95% confidence intervals. B: Coefficients of the regression. Labels report unstandardised effect size, $t$-value, and $p$-value. Error bars indicate $t$-adjusted, 95% Gaussian confidence intervals.

data (($\chi^2(4) = 19.70$, $p < .001$, adjusted $R^2 = .241$). In support of the compliance hypothesis, we find that the weights for the main effect of Group and Individual conditions are significant, whereas that of the Computer condition is not (Fig 5B; Baseline: $\beta = .170[-1.02, 1.36]$, $t = .279$, $p = .780$; Computer: $\beta = .176[-1.50, 1.85]$, $t = .206$, $p = .837$; Individual: $\beta = 3.69[1.89, 5.50]$, $t = 4.011$, $p < .001$; Group: $\beta = 4.67[3.29, 6.03]$, $t = 6.67$, $p < .001$). Furthermore, the interaction term with compliance is only significant in the Computer condition ($\beta = 25.31[14.75, 35.87]$, $t = 4.70$, $p < .001$) and not in the other experimental conditions (all $p > .05$). These findings suggest that attitude convergence in the Computer condition is mainly driven by experimenter compliance, rather than contagion.

**Preference learning.** If compliance could explain attitude convergence in the Computer condition, and given that we tend to exclude the presence of contagion in our experiment, attitude convergence in the Group and Individual conditions could be explained instead by either the preference learning or the norm learning hypotheses. We first test the second prediction of the preference learning hypothesis, namely that learning about others' attitude should significantly increase participants' consistency. We thus test whether choice consistency increased, and if this increase is higher after observing a human agent (Individual, Group conditions) than after predicting a computer's choices or nothing at all (Computer and Baseline conditions). Shapiro-Wilk test for normality is significant (all $p < .001$), therefore we adopt nonparametric tests. Wilcoxon signed-rank tests show that $\sigma_{after}$ was indeed significantly smaller than $\sigma_{before}$ in all conditions ($p < .001$; Fig 4B).

When we compare consistency increase across conditions, we find that participants become more consistent in the Group condition than in the Baseline condition ($z = 2.85$, $p = .026$, $r = .20[.06, .33]$, $BF_{10} = 6.83$; Kruskal-Wallis test: $\chi^2(3) = 8.50$, $p = .037$, $\varepsilon^2 = .02[0, .07]$). Despite this significant result, Bayes Factor analyses for most comparisons favour the null hypothesis (no differential increase; Baseline-Computer: $BF_{01} = 6.07$; Baseline-Individual: $BF_{01} = 4.48$; Computer-Individual: $BF_{01} = 4.53$; Individual-Group: $BF_{01} = 3.44$). We question then that the preference learning hypothesis does adequately explain our data.

**Norm learning.** We use the data from the norm elicitation task to test the plausibility of the norm learning hypothesis and to distinguish between norm uncertainty and norm salience.

We first compare appropriateness ratings between prosocial and antisocial participants in the Computer condition using a series of Kruskal-Wallis tests (one test for each rating; Fig 6, top). Appropriateness ratings are statistically different for every rating, even after correcting for multiple comparisons (all $p < .004$). These ratings link norm perception to social attitude: prosocial participants seem to consider it very appropriate to give money to the other and very inappropriate to take money, while the opposite is true for antisocial participants. Thus, it appears that participants' social attitudes are influenced by normative beliefs.

Given this evidence in support of the Norm Learning hypothesis, we proceed to testing norm uncertainty. To check this, we test whether the distribution of appropriateness ratings differ across conditions by participant type, which would happen if norm uncertainty was present. Two Kruskal-Wallis tests out of twenty-four are statistically significant, but do not survive the correction for multiple comparisons (all $p > .153$). We also perform Bayes Factor analyses, but given the large number of pairwise comparisons we choose to adopt a wider Cauchy prior, with spread $r = \sqrt{2}$. Results show that the null hypothesis is favoured by 67 tests out of 72 (Fig 7). Similar ratings in human (Group/Individual) and Computer conditions are thus not compatible with the norm uncertainty hypothesis, leaving norm salience as the only available explanation. Thus, authority compliance and norm salience are the only hypotheses that we failed to reject.

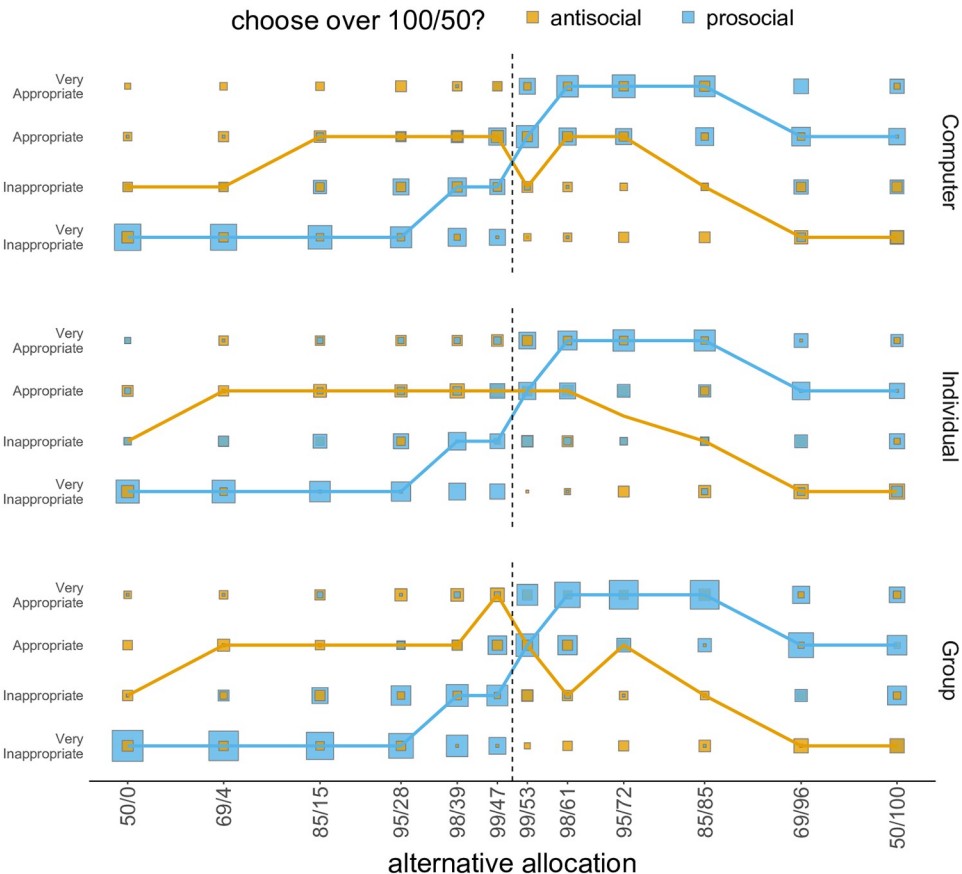

**Fig 6. Norm elicitation task.** Appropriateness ratings for prosocial and antisocial participants in the Computer (top), Individual (centre), and Group (bottom) conditions. Only participants below threshold are plotted. Square size is proportional to the number of participants, whereas the lines connect the median ratings for each alternative allocation.

## Discussion

### Drivers of social conformity

In this study, we identified and estimated the contributions of several competing explanations to attitude conformity in social decision making. Attitude conformity was assessed using a series of cognitive models coupled with several experimental conditions and complementary measures, which helped to test the predictions of these hypotheses. Participants' attitude became more prosocial or antisocial when they learned about the choices of an extremely prosocial or antisocial agent, regardless of whether the agent was a group of people, one person, or a computer. We found, however, that attitude compliance in the Computer condition was primarily driven by those participants who were more likely to conform to authority demands, suggesting that attitude change in this condition was primarily driven by compliance with the experimenter's expectations rather than conformity to the observed agent. Once we had accounted for authority compliance, computational modelling helped us to disentangle the surviving hypotheses, preference learning and norm learning. We first tested the prediction of the preference learning hypothesis that participants should have become more self-consistent in their choices after observing human agents. Since results disconfirmed this prediction, we proceeded to test one prediction of the norm learning hypothesis, namely that the behaviour

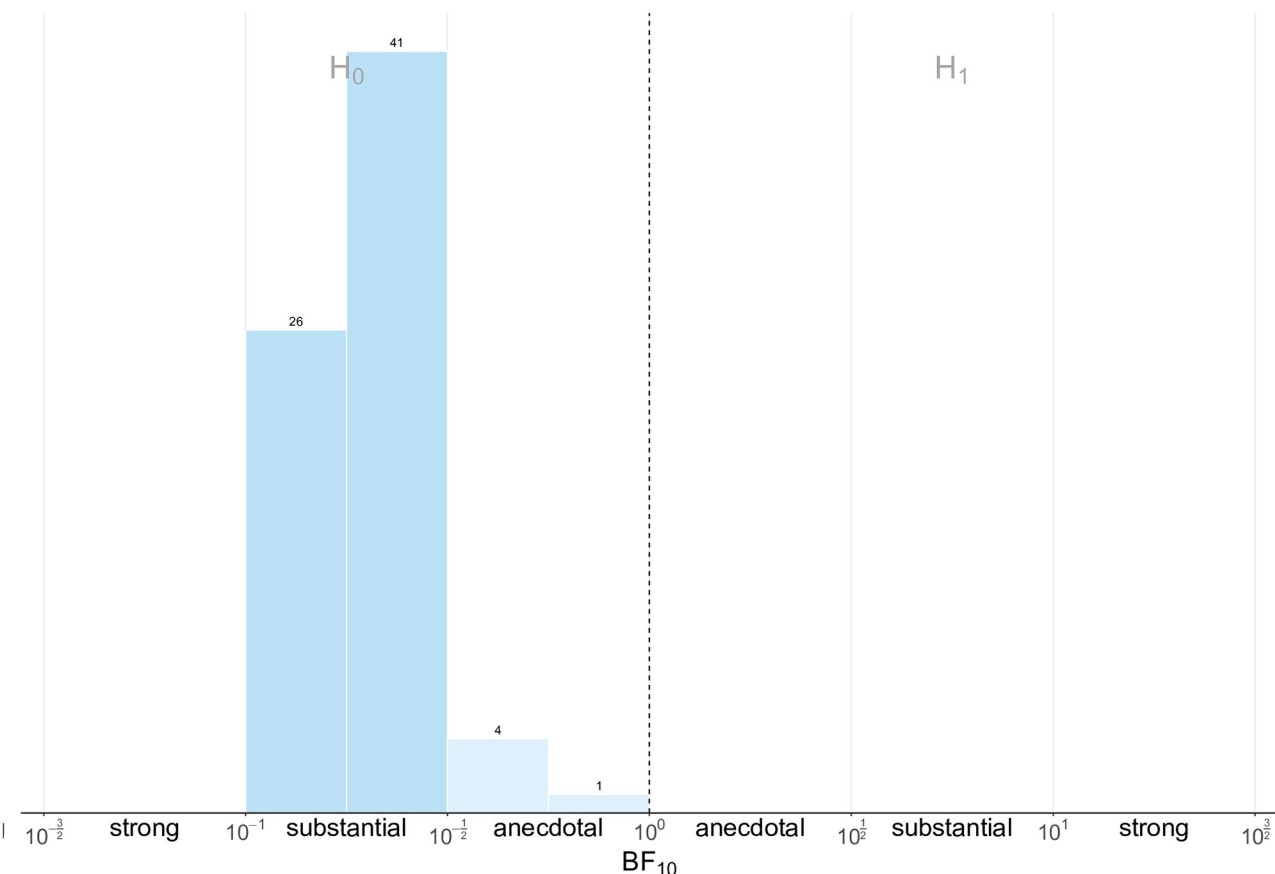

**Fig 7. Norm comparison between conditions.** Histogram of Bayes Factor values for the 72 pairwise comparisons of appropriateness ratings for each allocation, for each pair of experimental conditions, binned by strength of evidence. Evidence in favour of the null hypothesis ($H_0$) increases from right to left, and vice versa.

of participants in the game is reflected by their beliefs about what is considered appropriate or not. Results from a norm elicitation task [55] confirmed this prediction, adding evidence in favour of the hypothesis that participants conform mainly because of social expectations. As an exploratory analysis, we additionally tested the norm uncertainty hypothesis, which posits that participants are uncertain about the norm underlying the game and learn about it by observing other human agents. Results from the norm elicitation task however suggest that participants do not change their norm perceptions upon observing human agents. Based on this finding, we speculate that social conformity in the experiment occurs because participants learn *how salient* following the norm is (i.e. how unlikely it is for someone to deviate from the norm). Given the exploratory nature of this result, we cannot exclude other interpretations: a broader set of responses in the norm elicitation task paired with a properly powered study design should be able to assess what mechanisms underlie norm learning.

A number of findings support the idea that norms and the beliefs related to them are at the basis of social attitudes. Social appropriateness has been shown to play a role in decisions in various economic games [38, 55, 65–68]. More relevantly to our study, it was found that anonymity, and therefore reduced accountability, appears to have clear effect on allocation choices. Experiments with increased anonymity—also with respect to the experimenter, i.e. double blind paradigms—show plummeting contributions in economic games such as the

Dictator game [51, 69]. At the same time, even subtle cues of being observed seem to increase contributions [70] (although see [71] for a recent failed replication). The impact of reputation can also account for attitude change driven by compliance. Authority compliance is indeed a phenomenon analogous to conformity, as it links attitude change to vertical influences, as opposed to peer observation. Participants who have a strong tendency to choose the alternative option, regardless of whether it is beneficial or detrimental for the other and regardless of the identity of the observed agent, may think that this is what authority wants, and that this is the norm in the experiment [38, 72, 73]. The complementary result, that participants who are not influenced by authority only change their attitude when learning about other humans' behaviour, works in a similar fashion. By learning to predict the agent's behaviour, participants deduce how salient following the norm is for others, and change their behaviour to be more consistent with them. Therefore, we can conclude that the two effects that we observe—authority compliance and attitude conformity in human conditions—are both in line with the general social norms explanation.

The results of this study prompt some additional thoughts about the process of learning social norms. First, we observe that information about norms can spread through indirect transmission [74]. During the experiment participants cannot interact in any way with the observed human agent—who is not physically present—but participants can nevertheless extract some information about the norm from the observed behaviour in the manipulation phase. Indirect transmission thus highlights how adherence to social norms can be pervasive in dispersed and loosely regulated groups such as online communities. Second, the fact that participants conform by learning how salient a norm is implicates that if a norm is already salient among a group of individuals then such group should be more resilient to conformity influences. If future studies do confirm that norm perception prior to observation does predict conformity, this could suggest new measures to countervail polarisation in social discourse.

Our contribution not only fosters and provides better characterisation of the norm learning hypothesis, but also systematically devalues the several competing explanations that we tested, that to our knowledge were not yet properly compared in one framework. These *non-social* hypotheses include time-dependence, contagion, and preference learning. The social/non-social distinction is crucial here as it gives an insight into how to interpret conformity dynamics in interpersonal relations: if a person changes her attitude we suggest that this change has to be primarily social in nature, and linked to the changes in social context in which the decision maker is placed. This idea can have profound implications for studying any social learning mechanisms and social decision making in general. Specifically, many non-social explanations of the change in behaviour can be ruled out.

Whereas the experimental design focused on some of the most prominent hypotheses in the literature, it is also possible that further mechanisms may guide participants' social conformity. Social mimicry for instance suggests that participants change attitude in order to increase ties within a group [14, 75, 76]. The need to belong [77] could represent an alternative explanation to norm compliance, although it is not able to explain why prosocial and antisocial participants display different appropriateness beliefs. It is also by all means possible that several mechanisms contribute together to the present results; our main conclusion is that they probably operate on a social level. Concurrently, we do not claim that results on social behaviour directly apply to other domains of decision-making, such as risk or temporal preferences. Our research method, however, could be applied to other types of preferences to test whether these results extend to other types of choices.

## Cognitive modelling

Cognitive modelling does not only play a fundamental role in testing the predictions of the different hypotheses, but is also inherently connected to two additional contributions of this paper. First, we add to the series of studies challenging the conceptualisation of preference as a stable trait of people, and thus the use of the softmax function as the privileged method to model value-based choices. Studies on both risk [78, 79] and inter-temporal preferences [23, 80] have in fact highlighted how choice variability can be better explained by fluctuations of subjective preferences rather than "errors" in comparing different alternatives. This is in line with our finding that the Variable Attitude model explains behavioural data better than the Stable Attitude model. While we do not claim that computational distortions are absent during the estimation of value, we nonetheless support the idea that this mechanism cannot be the only one, nor can it be the main cause for choice inconsistencies in value-based decision making.

This interpretation finds additional support in recent perspectives on brain architecture, which hold that value representation is less specifically defined and is more distributed than current thinking suggests [81–84]. Assuming that preferences vary across contexts and across time requires a network of resources that not only keeps track of the current internal state, but that takes also into account the situational factors and the different scopes within which the choice is considered. For instance, a decision to act prosocially would require the integration of the tendency of an individual to help others, considerations related to the nature of the interaction and of the other person, the general goals of the decision maker, as well as the history of choices preceding that particular choice. Considering the complexity of a choice and of the neural substrates that make it possible, it seems hard to postulate the stability of subjective value as a justifiable premise for studying personal preferences and attitudes.

While we stand by the current findings, future research could improve the Variable Attitude model by accounting for some of its limitations. One way to do this could be to integrate both types of choice variability (errors in comparison and variability in attitude) under a common cognitive model to test whether these mechanisms co-exist and what are their individual contributions (see for example [85–88]). Such a model, however, requires either a prohibitive number of trials per participant, or the integration of some other type of information. This problem could possibly be overcome by integrating temporal information to simple choice data: several studies have successfully analysed subjective choices with this method before using so-called sequential sampling models (SSM, see for instance [21, 89]). While this approach would require challenging improvements, such as disentangling variability both within and between trials, it could also promote the analysis of other decision components, such as the trade-off between fidelity with one's preferences and speed in making a decision.

A second contribution of cognitive modelling is the use of a computational parameter to directly measure the impact of authority compliance on the decision process (Bias parameter $\kappa$ and Compliance and $\kappa$ relation). This parameter correlates with the compliance index that we used in the present study to categorise participants. We propose that this parameter can be used independently to measure compliance to authority demands. Directly including the effect of compliance in the computational model has the advantage that other estimates, such as the person's attitude or its choice consistency, are corrected for the presence of this effect. We also consider this estimation procedure as more reliable than alternatives in the literature: while other methods indeed exist, they are based on ad hoc tasks to quantify authority demand (e.g., [50, 90]), whereas the measures we use work within the main task of the experiment, thus reducing the risk that results in one task do not extend to another. As a limitation of our approach, it could be argued that using a default option might seem too unequivocal; we argue

however that this feature of the task design actually simplifies the expression of attitude by participants as it makes value comparison less challenging also from a computational point of view (see for instance [91, 92]). We thus think that our computational parameter could be of value to researchers who need to control for the influence of the experimenter when fitting decision models.

## Limitations

Our study does not come without some limitations. The experimental design is between-subjects, and it is thus not possible to compare directly the effect of the various manipulations, nor does it allow to exclude the possibility that multiple mechanisms are at work simultaneously. While this weakness does not fundamentally challenge the reported findings, implementing an intermixed design such as the ones proposed in [18, 27] or [19] could yield more powerful predictions and interpretations. A second constraint of our experiment design is that in some conditions we could not reach the pre-determined sample size necessary to achieve the power $1-\beta = .95$. We note however that our findings seem robust, even when applying design changes such as different ways to account for the influence of compliance (S1 Table) or changing the dependent variable (S6 Analyses), suggesting that this problem might be not too concerning.

Another limitation of the design is that, given that the attitude of the observed agent was fixed, social distance from the agent and attitude change are correlated. This means that we may be missing a connection between how close one's initial attitude is to the observed agent's and how much she will conform after learning. Indeed, recent research suggests that similarity with the observed agent influences the effect of conformity [21]. To solve this problem, in future experiments we propose to dynamically adjust the attitude of the agent depending on participants' own attitude. This design can also help to understand what happens when prosocial participants observe an antisocial agent and vice versa. We have deliberately excluded this question from consideration in our experiment because we were not sure *ex ante* if we would manage to separate the effect of learning about a very socially distant agent from the drift of attitudes towards selfishness (though, *ex post* we know that such time-dependence is not a likely explanation of the findings, which should make it straightforward to test hypotheses about observing others with very different attitudes). Previous results suggest however that, at least concerning antisocial participants, cross-type conformity should be less pronounced than same-type conformity [26].

We would like to note that, contrary to the predictions of the norm learning hypothesis, attitude change in the Individual condition was not significantly smaller than attitude change in the Group condition. This unexpected result could be linked to the fact that participants were not informed about the size of the group, which in turn could have influenced their representation of the agent. A key direction for the future research will be to explore the relationship between group size and attitude change (e.g., [93]). Another possible explanation for the lack of the difference between the Individual and Group conditions could be that participants in the Group condition were not connected in any way with the group of people whose behaviour they observed, and that they would have conformed more on average had they identified more with the group. This scenario could be compatible with recent findings suggesting that norms are stronger when there is a stronger group identification [27, 94]. Testing this idea would require more rigorous control of the perception of the group by participants.

Finally, we would like to comment on the implicit assumption that we make when analysing responses in the norm elicitation task. Specifically, we assume that the norms elicited in the Computer condition were not influenced by the predicting of computer's behaviour in the second task, and thus these norms are those that participants had in mind while choosing in

the first part of the resource-allocation game. It can be argued that learning about the computer's "attitude" can change the perception of norms and that our assumption is therefore incorrect. We disagree with this opinion on the following grounds. The computer is not a social agent, so whatever it is doing should not, by definition, change the perception of the *social* environment that participants are in. This is evident from the fact that attitude conformity is not significant in the Computer condition after regressing compliance to authority against attitude convergence. Moreover, in a post-experimental questionnaire, 66 of the 74 participants in the Computer condition reported that, in their opinion, the computer was acting either according to a mathematical rule (e.g., addition or subtraction between the players' payoffs; N = 62), or randomly (N = 4). It is thus unlikely that these participants 'humanised' the choices made by the computer agent. The fact that we do not find any differences in the elicited norms across the three conditions is much more likely to reflect the stability of normative beliefs across conditions rather than beliefs that change in all three conditions in exactly the same way. It can nonetheless still be argued that the mere experience of the task influences norm perception. This idea has been directly tested by [95] who found no evidence to support it. Overall, we believe therefore that our treatment of norms in the Computer condition is legitimate.

## Conclusion

In conclusion, we find that compliance to authority and learning how consistently others follow social norms are the most likely explanations behind prosocial and antisocial conformity. We hope that these findings will shed some light on the polarisation and viral diffusion of information online, that it will push towards a similar systematic exploration of preferences across other domains, and to a renewed interest in the cognitive and brain processes underlying these changes.

## Supporting information

**S1 Fig. Starting parameters' values.** Individual parameters' distribution before manipulation, by condition. Each vertical line represents a participant, jittered for illustration purposes. Parameter $\alpha$ represents participants' estimated social attitude; $\sigma$ estimates participants consistency across choices, $\kappa$ indicates the penalty points of the default allocation with respect to the alternative allocations; $\varepsilon$ indicates the percentage of trials in which there was likely a response mistake by the participant.
(TIF)

**S1 File. List of allocations.** An.xlsx file containing all the allocations used in the task: osf.io/ y8v63.
(XLSX)

**S1 Methods. Preregistration amendments.**
(PDF)

**S1 Table. Varying the compliance index.**
(PDF)

**S1 Analyses. Model identification.**
(PDF)

**S2 Analyses. Attitude convergence $\delta_{\mathrm{diff}}$ parameter recovery.**
(PDF)

**S3 Analyses. Balance between conditions.**
(PDF)

**S4 Analyses. Attitude distance from the agent before prediction.**
(PDF)

**S5 Analyses. Time dependence analyses.**
(PDF)

**S6 Analyses. Attitude polarisation.**
(PDF)

**S7 Analyses. Attitude convergence and consistency increase.**
(PDF)

## Acknowledgments

We would like to thank Dimitris Katsimpokis, Stephan Nebe, and the members of the Learning and Decision Making group at University of Trento for invaluable comments. All mistakes are our own.

## Author Contributions

**Conceptualization:** Folco Panizza, Alexander Vostroknutov, Giorgio Coricelli.

**Data curation:** Folco Panizza.

**Formal analysis:** Folco Panizza.

**Funding acquisition:** Giorgio Coricelli.

**Investigation:** Folco Panizza.

**Methodology:** Folco Panizza, Alexander Vostroknutov.

**Project administration:** Giorgio Coricelli.

**Software:** Folco Panizza.

**Supervision:** Alexander Vostroknutov, Giorgio Coricelli.

**Visualization:** Folco Panizza.

**Writing – original draft:** Folco Panizza.

**Writing – review & editing:** Folco Panizza, Alexander Vostroknutov.

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
