## [Decision Letter · Decision Letter 0]

2 Jul 2021

Dear Dr. Panizza,

Thank you very much for submitting your manuscript "How Conformity Can Lead to Polarised Social Behaviour" for consideration at PLOS Computational Biology. As with all papers reviewed by the journal, your manuscript was reviewed by members of the editorial board and by two independent reviewers. The reviewers appreciated the attention to an important topic. Based on the reviews, we are likely to accept this manuscript for publication, providing that you modify the manuscript according to the review recommendations.

Sincerely,

Jean Daunizeau

Associate Editor

PLOS Computational Biology

Natalia Komarova

Deputy Editor

PLOS Computational Biology

[LINK]

Reviewer's Responses to Questions

**Comments to the Authors:**

Reviewer #1: The present study investigated the social conformity of other-regarding preference in humans. Specifically, the authors examined conformity in the domain of social behaviour and tested the underlying psychological mechanism by combining behavioural experiments with computational modelling. While we know that people exhibit social conformity, its computational basis remains elusive. In this sense, I believe this study will advance our understanding of human social conformity. I have reviewed the manuscript for another journal, and the authors have already addressed most of my concerns. My comments are therefore relatively minor.

Does contagion or automatic imitation imply conformity occur regardless of whether the observed agent is human or non-human? How about human-specific automatic imitation reported in psychological literature?

Are the groups of participants matched in terms of sex, age, baseline social preference and other characteristics? In between-participant design, this issue should be carefully checked.

Reviewer #2: In this manuscript, Panizza and colleagues investigate how learning social behaviour of others influences one’s own social attitudes. They propose to disentangle 6 different hypothesis to account for the past empirical observations of such attitude alignment: Time-dependence, Contagion, Compliance, Preference learning, Norm uncertainty and Norm salience. To tease apart those hypotheses, they leverage a between-subjects experimental design of a “standard” attitude alignment experiment (own-preference elicitation – observation/prediction of other preferences - own-preference elicitation) with different treatments, which are defined by a manipulation of the observation/prediction phase: baseline (no observation), computer (predicting a computer behavior), individual (predicting preferences of one individual) and group (predicting preferences of a group of individuals). The different hypotheses make different qualitative prediction wrt to attitude alignment in the different treatments.

There is a lot to like in the manuscript: the topic is clearly interesting and important, the authors put a real effort to propose an exhaustive test of multiple credible hypotheses, the design is clever/elegant and suited to address the question(s) at hand, the manuscript features a very transparent and exhaustive reporting of the results (with Bayesian statistics), a solid preregistration (with transparent reporting of deviations from original plan) and open research practices.

I only have a few suggestions that I hope the authors will find useful to improve the manuscript.

First, I find the modelling part quite under-exploited which I suspect is due to the fact that the manuscript was initially formatted for another journal. For PLoScb, I suggest a stronger focus on the modelling approach, with modelling methods (Supp. Mat. A7) and modelling results (Supp. Mat. B1) incorporated in the main text. Also, because a lot of analyses apply to attitude alignment (d_diff), which is a model parameter, the other would need to provide a parameter recovery exercise (see e.g. Wilson and Collins, eLife 2019). A model identification would also be needed to support the model comparison results.

To provide a simple, first overview of the main effect of interest, I suggest that the authors also include a Figure (before the current Figure 2) which simply display the attitude convergence in the different conditions

Some effects size/direction are missing (e.g. paragraph about attitude convergence differences between conditions p.9)

Fig 3B : Unless I’m missing something, there seems to be an inconsistency between the significance stars and the CI (and the results reported in the Main text)

P9: Paragraph head “consistency increase” should be “preference learning” for clarity/consistency

**Have the authors made all data and (if applicable) computational code underlying the findings in their manuscript fully available?**

Reviewer #1: Yes

Reviewer #2: None

PLOS authors have the option to publish the peer review history of their article (what does this mean?). If published, this will include your full peer review and any attached files.

Reviewer #1: No

Reviewer #2: No

Figure Files:

Data Requirements:

Reproducibility:

References:

---

## [Decision Letter · Decision Letter 1]

2 Sep 2021

Dear Dr. Panizza,

Thank you very much for submitting your manuscript "How Conformity Can Lead to Polarised Social Behaviour" for consideration at PLOS Computational Biology. As with all papers reviewed by the journal, your manuscript was reviewed by members of the editorial board and by several independent reviewers. The reviewers appreciated the attention to an important topic. Based on the reviews, we are likely to accept this manuscript for publication, providing that you modify the manuscript according to the review recommendations.

Sincerely,

Jean Daunizeau

Associate Editor

PLOS Computational Biology

Natalia Komarova

Deputy Editor

PLOS Computational Biology

[LINK]

Reviewer's Responses to Questions

**Comments to the Authors:**

Reviewer #1: The authors have adequately address all the concerns.

Reviewer #2: It seems that the authors have satisfactorily addressed my previous comments. I am therefore happy to recommend the manuscript for publication, conditional on the minor issue/question below being addressed as well. Congratulations to the authors for this very fine piece of work.

Minor issue:

I am surprised about the reported effect-size confidence intervals and how they relate to p-values e.g. lines 521-522: “Baseline and Group conditions are still significantly different (z = 5:20, p < :001, 520 r = -.37[-.51; .24], BF10 = 2117:44), and so are Baseline and Individual conditions (z = 3:90, p = :001, r = -:31[-:45; :15], BF10 = 54:87).” Aren’t some “minus” signs missing from the second terms in the brackets (to exclude 0 from the CI and reach high significance levels of P<0.001)? Also would better correspond to the reported mean the center of the CI. Please check throughout the manuscript.

**Have the authors made all data and (if applicable) computational code underlying the findings in their manuscript fully available?**

Reviewer #1: Yes

Reviewer #2: None

PLOS authors have the option to publish the peer review history of their article (what does this mean?). If published, this will include your full peer review and any attached files.

Reviewer #1: No

Reviewer #2: No

Figure Files:

Data Requirements:

Reproducibility:

References:

---

## [Editor Report · Decision Letter 2]

5 Oct 2021

Dear Dr. Panizza,

We are pleased to inform you that your manuscript 'How Conformity Can Lead to Polarised Social Behaviour' has been provisionally accepted for publication in PLOS Computational Biology.

Best regards,

Jean Daunizeau

Associate Editor

PLOS Computational Biology

Natalia Komarova

Deputy Editor

PLOS Computational Biology

---

## [Editor Report · Acceptance letter]

15 Oct 2021

PCOMPBIOL-D-21-00918R2 

How Conformity Can Lead to Polarised Social Behaviour

Dear Dr Panizza,

I am pleased to inform you that your manuscript has been formally accepted for publication in PLOS Computational Biology. Your manuscript is now with our production department and you will be notified of the publication date in due course.

With kind regards,

Katalin Szabo
